# Mental health services in response to the COVID-19 pandemic in high-income countries: protocol for a rapid review

Ge Yu [1,2] Dawn Craig,[1,2] Yu Fu [1,2]

[1]NIHR Applied Research Collaborative North East and North Cumbria, Newcastle upon Tyne, UK
[2]Population Health Sciences Institute, Newcastle University, Newcastle upon Tyne, UK

**Correspondence to**
Dr Ge Yu; ge.yu@ncl.ac.uk

## ABSTRACT

**Introduction** The COVID-19 pandemic has caused disruptions to mental health services, forcing the rapid implementation of alternative ways of delivering services alongside a greater immediate, and continuously growing, demand across those services. The care and level of mental health service provided are felt to be inadequate to respond to the increasing demand for mental health conditions in the time of the pandemic, leading to an urgent need to learn from service change and consequences to inform solutions and plans to support the NHS postpandemic plan in the UK. This rapid review aims to understand the changes in mental health services during the pandemic and summarise the impact of these changes on the health outcomes of people with mental health conditions.

**Methods and analysis** Cochrane CENTRAL, MEDLINE, Embase and PsycInfo will be searched for eligible studies with key terms indicating mental health AND COVID-19 AND health services. Peer-reviewed empirical studies aiming to investigate or describe new models of care, services, initiatives or programmes developed or evolved for patients (aged 18 years or over) with mental health in response to COVID-19, published in the English language and undertaken in a high-income country defined by Organisation for Economic Co-operation and Development (OECD) member will be included. Studies reporting views of the general public, letters of opinion to peer-review journals, editorial or commentaries will be excluded. Study selection and data extraction will be undertaken independently by two reviewers. Evidence will be summarised narratively and in a logic model.

**Ethics and dissemination** Ethics approval is not required for this review. A list of interventions/services/models of care delivered to people with mental health conditions will be grouped as 'Do', 'Don't' and 'Don't know' based on the evidence on effectiveness and acceptability. The results will be written for publication in an open-access peer-reviewed journal and disseminated to the public and patients, clinicians, commissioners, funders and academic conferences.

**PROSPERO registration number** CRD42022306923.

## STRENGTHS AND LIMITATIONS OF THIS STUDY

⇒ This is a rapid review with a systematic search of literature on mental health services since the WHO declared the global outbreak of the COVID-19 pandemic on 11 March 2020.
⇒ This review will provide a rapid but robust collation of evidence in response to requests for timely evidence syntheses for decision-making purpose for the postpandemic period.
⇒ Outlining the objectives and methodology a priori will improve both transparency and quality and help reduce bias and enhance the reproducibility of the results.
⇒ Some limitations to the study design include studies limited to Organisation for Economic Co-operation and Development settings, exclusion of non-English studies, publication bias, quality of data, selection bias and no quality assessment in the rapid evidence review.
⇒ To help mitigate the limitations of the proposed study design, studies will be screened independently by two reviewers with a third reviewer consulted when there is a lack of consensus.

## INTRODUCTION

There was a substantial deterioration in mental health, and the prevalence of mental health symptoms increased both in previously healthy people and those with pre-existing mental health conditions since the outbreak of COVID-19.[1–3] It is estimated that the pandemic will lead to new or additional mental health support for up to 10 million people in England (around 20% of the population).[4] People with existing psychiatric diagnoses have reported increased symptoms and poorer access to services and support leading to relapse and suicidal behaviour.[1] While the increasing demand for mental health support/treatment inevitably exceeded the capacity of essential mental health services, the pandemic has significantly interrupted usual practice in the UK and worldwide. On 23 March 2020, a national lockdown was announced by the UK government with the public instructed to stay at home, socially distance and self-isolate with strict guidance about movement outside of one's household. The adaptations required to enable the delivery of mental healthcare services during this period of extended infection control

measures could have been disproportionally detrimental to those now living with mental health conditions (ie, autism,[5] obsessive compulsive disorder,[6] substance use disorder,[7] etc). Difficulties attending review appointments in person and closure of support services are likely to have impacted all those in, or in need of, active treatment.[8] The unequal impact of the pandemic and countrywide lockdown is likely to further entrench and exacerbate the existing structural inequalities in mental health among people with pre-existing mental health conditions before COVID-19. Furthermore, the mental health services provided have failed to meet the increasing demand for mental health treatment/support during the time of the COVID-19 pandemic.[9]

The UK National Health Service (NHS) has set up a long-term plan to improve mental healthcare services that are widely regarded as being under-resourced.[10] However, for people with mental health conditions, there is an incomplete picture of the impact of the pandemic on the pattern of mental health services. Despite bringing current service inadequacies to the forefront, the pandemic could provide an opportunity to rethink conventional approaches to mental health services planning to meet patients' needs. For example, remote community treatment and support has long been suggested but has not previously been implemented widely because of barriers and challenges from both healthcare staff and service users. Since the onset of the pandemic, the situation has changed.[11] Similarly, the threshold for hospital admission for mental illness varies between individuals and requires continuous adaptation over time. Therefore, learning from health service changes throughout the pandemic, and their consequences for people's physical and mental health is vital to inform practical policy solutions for integrated service recovery and effectively plan services that reach those with the greatest need.

## THE AIM OF REVIEW

The overall aim of this review is to: (1) identify changes in mental health services for adult patients in response to the pandemic and (2) understand the impact of the changes on their health outcomes in high-income countries.

## METHODS AND ANALYSIS

A rapid review is defined as 'a form of knowledge synthesis that accelerates the process of conducting a traditional systematic review through streamlining or omitting various methods to produce evidence for stakeholders in a resource-efficient manner'.[12] The WHO recommends rapid review methods as an efficient approach to provide rapid but relevant and contextualised evidence to the health decision makers when there are time, resources or other logistical constraints.[13–18] With rapid changes in service provision in response to the pandemic, a rapid review will be undertaken in a timely manner to provide an evidence base supporting the recommendation of

mental health services since COVID-19 and identify areas where the evidence base is lacking, and future research is required. The review will be guided by the Cochrane guidance for rapid reviews.[12] Preferred Reporting Items for Systematic Reviews and Meta-Analyses extension for Rapid Reviews guidance[19] will be followed for reporting. The review will be carried out between May 2022 and August 2022.

This protocol has been developed in advance of the review to improve the transparency and quality of the methods to help reduce bias and enhance the reproducibility of the results. This has been registered with the PROSPECT CRD42022306923.

## ELIGIBILITY

### Type of studies

Peer-reviewed quantitative, qualitative or mixed methods empirical studies aiming to investigate or describe new models of care, services, initiatives or programmes developed or evolved for patients with mental health in response to COVID-19 will be included. In addition, studies describing or comparing the setting, problems addressed, resource requirements, aim, service components, provider, method of delivery, objective and subjective effects of changes to mental health services in response to COVID-19 will be included. Studies reporting views of the general public, letter of opinion to peer-review journals, editorial or commentaries will be excluded.

### Type of participants

People aged 18 years or over experiencing mental health conditions as described by NHS[20] who were in need of mental health support during the pandemic.

### Type of health services

Interventions, services and models of care delivered in response to COVID-19 to provide support for adults with mental health conditions will be included.

### Type of outcome measures

Primary outcomes are objective measures and subjective effects of changes, efficacy or use of a service by mental health patients. Secondary outcomes are changes in knowledge, attitudes or satisfaction of service users and/or professionals and health inequalities.

### Type of study settings

According to level of economic development of the countries or regions under study, we will use membership of the Organisation for Economic Co-operation and Development (OECD)[21] as a reference 'cut-off' point to include high-income countries[22 23] to ensure a degree of similarities in the social security system, health system and socioeconomic and demographic characteristics as in the UK.

### Search methods

Cochrane CENTRAL, MEDLINE, Embase and PsycInfowill be searched for from 2019 to the present. A search

strategy has been developed for MEDLINE with support from an independent information specialist, using a range of keywords and subject headings representing COVID-19, mental health and low-income and middle-income countries (see online supplemental appendix). This will be used to inform the detailed search strategy for other databases. Reference lists and citation indexes of relevant studies will also be examined. Only OECD studies published in or after 2019 and in the English language (no resource available for translation) will be searched.

## Selection of studies

Studies identified from databases will be exported to EndNote X9[24] for deduplication. Study titles of abstracts will be screened independently according to the selection criteria. Any results that are inconclusive at the initial screen will be included and considered at full-text screening. All full-text papers will be screened independently by two researchers (GY and YF). Any discrepancies will be resolved by discussion and consensus. Where there is a disagreement between two reviewers, a third researcher (DC) will be consulted to reach a consensus.

## Data extraction

A data extraction sheet will be designed to capture information relating to new models of care, services, initiatives or programmes developed or evolved for patients with mental health problems in response to the pandemic. Data extracted will include author's first name, publication date, setting, study design, sample size, mental health conditions, characteristics of participants, service components, service provider, method of delivery, resources required, outcome measures and main study results. GY will extract all the data. YF will check for accuracy and completeness through random double-extraction of 10% of included studies. Where a study appears to have multiple citations, original authors will be contacted for clarification. All information from multiple citations will be used if no replies are received.

## Quality assessment and quality control

Quality assessment will not be conducted. Instead, a tabulated and narrative synthesis will be undertaken to report the results of included studies and discuss reasons for differences in this rapid review as suggested by current best practice guidance.[13 25 26]

The following steps will be taken to ensure quality control for the searching, screening, data extraction and coding process. GY will conduct screening and data extraction following predetermined inclusion criteria and data extraction framework. For articles that are retrieved and full text saved, YF will check 10% of the coding to ensure they meet the screening criteria. Where there is a disagreement between two reviewers, a third researcher (DC) will be consulted to reach a consensus. Synthesis of each outcome will be conducted by GY and independently revised by YF.

## Data synthesis and analysis

A tabulated and narrative synthesis of the results will be undertaken following current best practice[25–27] to conduct synthesis systematically and transparently. It will focus on the mental health services, mechanisms and their impact on health outcomes. A logic model will be produced to present context, service provision and outcomes. Possible unintended adverse outcomes will also be reported. Also, a list of interventions/services/models of care delivered to people with mental health conditions will be grouped as 'Do', 'Don't' and 'Don't know' based on the strength of the evidence on effectiveness and acceptability.

If data are available, outcomes of studies will be synthesised according to characteristics of study participants, for example, deprived communities, ethnic minorities, to produce evidence on health inequalities that is likely to have been exacerbated during the pandemic.

## Patient and public involvement

This study has been designed and developed in consultation with two public members (one with lived experience) to ensure their input on the study design. They both read and commented on the review summary, search strategies, eligibility and plans to synthesise data and dissemination strategies. They valued the potential impact of this review on NHS plans for mental health postpandemic. It has been agreed that the process of this rapid review will be presented to both members for their further comments.

## ETHICS AND DISSEMINATION

As this rapid review will only consider published literature, no ethics approval is needed. Dissemination will be led by the research team and supported by the public member and the wider project advisory group. Results of this review will contribute to reports that will be produced and shared with the National Institute for Health and Care Research (NIHR) Three Research Schools and NIHR Applied Research Collaboration North East and North Cumbria. The findings will be published in peer-reviewed journals, and a plain study summary will be disseminated to people receiving mental healthcare, groups and forum that the project public members, practitioners and commissioners are connected. An abstract will be prepared for academic conferences such as the Society for Academic Primary Care Annual Conference.

**Contributors** GY and YF conceived the study idea and design. GY drafted the initial manuscript, and GY, YF, and DC reviewed the manuscript and provided input to the final version.

**Funding** This publication presents independent research funded by the National Institute of Health and Care Research (NIHR) School for Primary Care Research as part of the Three NIHR Research Schools Mental Health Programme (award number MHF018 and MH027) and also supported by the NIHR (Applied Research Collaboration North East and North Cumbria (NIHR200173)). The views expressed are those of the authors and not necessarily those of the NIHR or the Department of Health and Social Care.

**Disclaimer** The views expressed are those of the authors and not necessarily those of the NIHR or the Department of Health and Social Care.

**Competing interests** None declared.

**Patient and public involvement** Patients and/or the public were involved in the design, or conduct, or reporting, or dissemination plans of this research. Refer to the Methods section for further details.

**Patient consent for publication** Not applicable.

**Provenance and peer review** Not commissioned; externally peer reviewed.

**ORCID iDs**
Ge Yu http://orcid.org/0000-0002-0891-2501
Yu Fu http://orcid.org/0000-0003-4972-0626

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
