## [Reviewer comments · BMJ Open]

ARTICLE DETAILS

TITLE (PROVISIONAL)	Mental health services in response to the COVID-19 pandemic in high-income countries: protocol for a rapid review
AUTHORS	Yu, Ge; Craig, Dawn; Fu, Yu

VERSION 1 – REVIEW

REVIEWER	Kourgiantakis, Toula University of Toronto, Factor-Inwentash Faculty of Social Work
REVIEW RETURNED	27-Mar-2022

GENERAL COMMENTS	This is such an important topic which is why I have provided some feedback on how to strengthen it. Abstract Your abstract introduction is well written and positions your problem and arguments better than your actual introduction of the paper. The method inclusion and exclusion criteria are unclear (see my comments below). Strengths and limitations Be more specific with the first statement and mention that it is since the start of the COVID-19 pandemic Do not use an abbreviation in your strengths and limitations that has not been written out in full first Your limitation states selection bias and this is important and should be a separate point where you mention it and also explain how you will mitigate this bias Introduction I would not introduce the country using the abbreviation at the outset. If it is used more than once, you can include the abbreviation in parentheses (UK), otherwise it should be written out in full (i.e., United Kingdom). I would change the first sentence as you are starting by saying how many people tested positive which we have learned is not necessarily that relevant. As you mention in your fourth sentence, it is the lockdowns and other coercive measures which created isolation and reduced mental health services, as well as fear caused by various sources that have worsened or caused mental health concerns. It is misleading and probably inaccurate to say that contracting the virus caused mental health concerns. I suggest re-writing your introduction with a different opening sentence and keeping in mind the aim of your review. If your aim is to show how mental health services were impacted then it is important to start with the fact there were increased needs during the pandemic caused by the measures which reduced social support, increased isolation, reduced ability to use already established coping strategies, increased fear etc. In addition, while the needs were higher during the pandemic, the services decreased. I would then focus a bit on how services shut down,
--

	pivoted to virtual, etc. There were inadequate services in the face of increased needs. Lastly, it is critical to mention how this affected specific populations and the inequities that widened as a result of measures that were more applicable to individuals who could work from home, who had financial means, who were able to assist children with remote learning, who had technology/language skills, who did not need specialized services (i.e., autism, OCD etc). I think it is important that you specify whether substance use is included and if it is, it should also be mentioned in your introduction. I recommend including substance use because we know that a high percentage of individuals with mental health concerns are also misusing substances. I think it would be helpful to have research objectives and/or research questions as stated in the rapid review recommendations by Garrity et al (2021). Method You should start with a definition of rapid reviews and Garrity et al (2021) define this type of review followed by why you chose to do this kind of review. You mention qual and quant studies but do not specify if you will include mixed methods. You include PICO, but the type of studies mentioned is very broad and unclear. You need to state the aim of the studies that you are including and be more specific about the ones you will exclude. It would be helpful to define or explain OECD countries for the reader. You provide a good reference, but understanding this classification system is important information for your protocol. For your data extraction, there does not seem to be a variable that extracts information about changes to mental health services. You write service components, method of delivery, etc., but that does not necessarily tell us how/if the services changed. It is important to re-phrase the statements noting that this is a rapid review therefore quality assessment is not needed. Instead, you should explain that you are conducting a narrative synthesis.
--	---

REVIEWER	Chakraborty, Nandini Leicestershire Partnership NHS Trust
REVIEW RETURNED	07-Apr-2022

GENERAL COMMENTS	1. 'Study selection and data extraction will be undertaken dependently by two reviewers'. This statement is unclear in the abstract. The details in the methodology further explains the roles of the two authors and also the role of a third reviewer in case of disagreement. I think the word dependently just needs to be dropped. 2. The review for appropriate reasons focuses on OCED countries. Trying to include all publications irrespective of country and language would make meaningful conclusions difficult as existing baseline for mental health services around the world vary widely. However, I think this should be reflected in the title of the review. Overall I the study addresses an important topic and is relevant to how NHS is structured, has responded and the way forward in treating a vulnerable group of patients.
--

VERSION 1 – AUTHOR RESPONSE

Reviewer 1 comments	Responses
Abstract Your abstract introduction is well written and positions your problem and arguments better than your actual introduction of the paper. The method inclusion and exclusion criteria are unclear (see my comments below).	This has been revised according to updated inclusion and exclusion criteria in the method section.
Strengths and limitations Be more specific with the first statement and mention that it is since the start of the COVID-19 pandemic Do not use an abbreviation in your strengths and limitations that has not been written out in full first Your limitation states selection bias and this is important and should be a separate point where you mention it and also explain how you will mitigate this bias	This section has been revised as suggested. The text has been changed to “This is a rapid review with a systematic search of the literature on mental health services since the WHO declared the global outbreak of the COVID-19 pandemic on 11th March, 2020.” Abbreviation has been replaced by the full name. “OECD” changed to “Organisation for Economic Co-operation and Development (OECD)”. Selection bias has been separated. More information has been added on mitigating the limitation.
Introduction I would not introduce the country using the abbreviation at the outset. If it is used more than once, you can include the abbreviation in parentheses (UK), otherwise it should be written out in full (i.e., United Kingdom). I would change the first sentence as you are starting by saying how many people tested positive which we have learned is not necessarily that relevant. As you mention in your fourth sentence, it is the lockdowns and other coercive measures which created isolation and reduced mental health services, as well as fear caused by various sources that have worsened or caused mental health concerns. It is misleading and probably inaccurate to say that contracting the virus caused mental health concerns. I suggest re-writing your introduction with a different opening sentence and keeping in mind the aim of your review. If your aim is to show how mental health services were impacted then it is important to start with the fact there were increased needs during the pandemic caused by the measures which reduced social support, increased isolation, reduced ability to	Abbreviation has been replaced by the full name. The introduction section has been re-written as suggested. It starts with the fact there were increased demands during the pandemic, followed by the interrupted provision of health services, and ends up with a discussion on the disproportionate impact of COVID on people with pre-existing mental health conditions.

use already established coping strategies, increased fear etc. In addition, while the needs were higher during the pandemic, the services decreased. I would then focus a bit on how services shut down, pivoted to virtual, etc. There were inadequate services in the face of increased needs. Lastly, it is critical to mention how this affected specific populations and the inequities that widened as a result of measures that were more applicable to individuals who could work from home, who had financial means, who were able to assist children with remote learning, who had technology/language skills, who did not need specialized services (i.e., autism, OCD etc). I think it is important that you specify whether substance use is included and if it is, it should also be mentioned in your introduction. I recommend including substance use because we know that a high percentage of individuals with mental health concerns are also misusing substances.

I think it would be helpful to have research objectives and/or research questions as stated in the rapid review recommendations by Garrity et al (2021).

The aim of this review has been added.

You should start with a definition of rapid reviews and Garrity et al (2021) define this type of review followed by why you chose to do this kind of review.

You mention qual and quant studies but do not specify if you will include mixed methods.

The definition has been added to the method section, followed by the rationale for choosing a rapid review design.

A rapid review is defined as “a form of knowledge synthesis that accelerates the process of conducting a traditional systematic review through streamlining or omitting various methods to produce evidence for stakeholders in a resource-efficient manner”. The WHO recommends rapid review methods as an efficient approach to provide rapid but relevant and contextualised evidence to the health decision-makers when there are time, resources or other logistical constraints. We will undertake a rapid review in a timely manner in response to urgent needs for an

	evidence base supporting the recommendation of mental health services since COVID-19 and to identify areas where the evidence base is lacking, and future research is required. This has now been revised. Peer-reviewed quantitative, qualitative, or mixed methods empirical studies will be included.
You include PICO, but the type of studies mentioned is very broad and unclear. You need to state the aim of the studies that you are including and be more specific about the ones you will exclude. It would be helpful to define or explain OECD countries for the reader. You provide a good reference, but understanding this classification system is important information for your protocol. For your data extraction, there does not seem to be a variable that extracts information about changes to mental health services. You write service components, method of delivery, etc., but that does not necessarily tell us how/if the services changed.	The type of studies section has been revised with more specific inclusion and exclusion criteria outlined. “Peer-reviewed quantitative, or qualitative, or mixed methods empirical studies aiming to describe new models of care, services, initiatives or programmes developed or evolved for patients with mental health in response to COVID will be included. In addition, studies describing or comparing the setting, problems addressed, resource requirements, aim, service components, provider, method of delivery, objective and subjective effects of changes to mental health services in response to COVID-19 will be included. Studies reporting views of the general public, letter of opinion to peer-review journals, editorial or commentaries will be excluded.” More information has been provided for the type of study settings. We changed text to “According to level of economic development of the countries or regions under study, we will use membership of the Organisation for Economic Co-operation and Development (OECD) as a reference “cut-off” point to include high-income countries, to ensure a degree of similarities in the social security system, health system and socioeconomic and demographic characteristics.” Data to be extracted will capture information relating to new models of care, services, initiatives or programmes developed or evolved for patients with mental health in response to COVID. This has been clarified in the Data extraction section. We have also revised the review title.

It is important to re-phrase the statements noting that this is a rapid review therefore quality assessment is not needed. Instead, you should explain that you are conducting a narrative synthesis.	This has been revised as suggested. The sentences “The tabulated and narrative synthesis will report the results of included studies and discuss reasons for differences in this rapid review as suggested by current best practice guidance.” was added.
---	---

Reviewer 2 comments	Responses
'Study selection and data extraction will be undertaken dependently by two reviewers'. This statement is unclear in the abstract. The details in the methodology further explains the roles of the two authors and also the role of a third reviewer in case of disagreement. I think the word dependently just needs to be dropped.	This error has been corrected.
The review for appropriate reasons focuses on OCED countries. Trying to include all publications irrespective of country and language would make meaningful conclusions difficult as existing baseline for mental health services around the world vary widely. However, I think this should be reflected in the title of the review.	Thanks for the suggestion. This review will focus on the OECD countries to ensure a degree of similarities in the social security system, health system and socioeconomic and demographic characteristics as in the UK. The title has been revised to also reflect the setting.
Overall I the study addresses an important topic and is relevant to how NHS is structured, has responded and the way forward in treating a vulnerable group of patients.	Thank you.

VERSION 2 – REVIEW

REVIEWER	Kourgiantakis, Toula University of Toronto, Factor-Inwentash Faculty of Social Work
REVIEW RETURNED	25-May-2022

GENERAL COMMENTS	The changes have strengthened your manuscript!
--